# Clinical and Biomechanical Progression after Ankle Joint Distraction in a Young Adolescent Patient with Haemophilia

**DOI:** 10.3390/ijerph182111405

**Published:** 2021-10-29

**Authors:** Nicolas Haelewijn, Sebastien Lobet, An Van Damme, Pierre-Louis Docquier, Maarten Eerdekens, Kevin Deschamps

**Affiliations:** 1Musculoskeletal Rehabilitation Research Group, Department of Rehabilitation Sciences, KU Leuven, Spoorwegstraat 12, B-8200 Brugge, Belgium; maarten_eerdekens@hotmail.com (M.E.); kevin.deschamps@kuleuven.be (K.D.); 2Service D’hématologie, Cliniques Universitaires Saint-Luc, Avenue Hippocrate 10, B-1200 Bruxelles, Belgium; sebastien.lobet@saintluc.uclouvain.be; 3Neuromusculoskeletal Lab (NMSK), Institut de Recherche Expérimentale et Clinique, Université Catholique de Louvain, Secteur des Sciences de la Santé, Avenue Mounier 53, B-1200 Brussels, Belgium; 4Secteur de Kinésithérapie, Cliniques Universitaires Saint-Luc, Avenue Hippocrate 10, B-1200 Brussels, Belgium; 5Service D’hématologie Pédiatrique, Cliniques Universitaires Saint-Luc, Avenue Hippocrate 10, B-1200 Bruxelles, Belgium; an.vandamme@uclouvain.be; 6Service D’orthopédie et de Traumatologie, Cliniques Universitaires Saint-Luc, Avenue Hippocrate 10, B-1200 Bruxelles, Belgium; pldocquier@gmail.com; 7Haute Ecole Leonard De Vinci, Institut D’Enseignement Supérieur Parnasse Deux-Alice, Division of Podiatry, Avenue E. Mounier 84, B-1200 Sint-Lambrechts-Woluwe, Belgium; 8Department of Podiatry, Artevelde University of Applied Sciences, Voetweg 66, B-9000 Ghent, Belgium

**Keywords:** ankle, arthropathy, haemophilia, joint distraction

## Abstract

Ankle joint distraction (AJD) has been described to be a valuable joint-sparing alternative to arthrodesis or arthroplasty; however, clinical endpoints associated to this surgical intervention are lacking. The current case report describes clinical and biomechanical outcome measures of ankle joint distraction in a 14-year-old patient with severe haemophilia A. Because of persistent and incapacitating pain and the poor response to conservative and invasive treatment options, ankle joint distraction was performed in this 14-year-old patient using an external fixator encompassing two Ilizarov full rings in the tibia and a foot ring fixed to the foot by four K-wires. State-of-the-art medical imaging and non-invasive skin marker-based 3D multi-segment foot modelling were performed in a pre- and post-operative stage. From a structural viewpoint, this AJD was a success since it improved and stabilised the osteo-cartilaginous lesions of the ankle. Biomechanical outcome measures associated with the 18-month follow-up were found to be suboptimal, showing an early plantarflexion pattern at the ankle joint during midstance and a tendency towards increased power absorption at the midfoot with peak power absorption being almost two times higher when compared to boys of the same age. From a functional viewpoint, we observed a clear reduction in the patients’ physical activities until one year after AJD. Despite these functional and structural improvements, recurrent painful phenomena, including the development of a complex regional pain syndrome (CRPS) and a stress fracture of the third metatarsal bone, were observed which are probably related with the development of recurrent subchondral oedema.

## 1. Introduction

Although regular factor replacement can reduce the incidence of joint bleeds and slow down the development of haemophilic arthropathy, the ankle joint remains particularly vulnerable even in children with haemophilia on primary or secondary prophylaxis and is now the primary joint affected [1,2,3]. In children with haemophilia, the ankle has more reported haemarthroses and worse joint scores compared with knees or elbows [1]. Recurrent hemarthrosis causes formation of hemosiderin and iron deposits, which induces inflammation related to reactive oxygen intermediates, abnormal neoangiogenesis, and catabolic cytokines [4]. Enzymatic degradation, synovial invasion, and inhibition of chondrocyte matrix synthesis are the main mechanisms responsible for the development of hemophilic arthropathy. The end stage of the disease is characterized mainly by the presence of muscle atrophy, inflammatory arthritis, abundant arthrofibrosis, low bone mineral density, and subchondral cysts [4]. In end stage disease, current treatment modalities include arthrodesis or arthroplasty. Ankle arthrodesis reduces the pain associated with motion at a degenerative joint. However, the procedure results in loss of range of motion and can negatively affect adjacent joints of the hind- and mid-foot. In an attempt to preserve motion and spare adjacent joint disease, ankle joint arthroplasties are now marketed and often compared to arthrodesis in terms of patient-reported and functional outcomes [5]. However, both of these common procedures are ablative to the native joint, an option which may not be desirable as a long-term solution in a young patient with ankle osteoarthritis. In such cases, a joint preservation procedure may be more desirable.

Ankle joint distraction can be a good alternative to spare the joint. Distraction arthroplasty does not use any graft material. This procedure employs an external fixator which crosses the ankle joint and applies a distraction force across the tibio-talar articulation (Figure 1) [6]. The theory behind distraction arthroplasty is that it allows for the reparative potential of the joint by removing mechanical stress. Ankle joint distraction has recently been used to treat three young patients with hemophilic arthropathy of the ankle, achieving clinical and radiological improvement at short-term follow-up, with no significant blood loss [7]. The authors considered this application justified by the noteworthy clinical outcomes of previous studies in ankle osteoarthritis. In a work by Ploegmaker et al., joint distraction achieved good and durable clinical results (73% of the cases) in post-traumatic ankle osteoarthritis at a minimum follow-up of seven years [8]. Distraction arthroplasty appears therefore to be a reasonable option to provide a good percentage of patients a satisfactory period of time of improvement, less invasive than “traditional” cartilage repair procedures using autograft or allograft.

However, evidence regarding the clinical, structural, and functional outcome in young patients and children is currently lacking. Moreover, in the few case reports about adults with haemophilia, functional outcome is mostly evaluated exclusively with clinical joint scores and patient reported outcome measures, whereas fundamental insight into the foot joint biomechanics as well as joint structure has not been objectively quantified.

In conclusion, current knowledge about both the clinical effects and complications of AJD in young patients with haemophilia is limited, while this is of utmost importance when striving for value-based healthcare. The objective of this case report was to report state-of-the art clinical, structural and biomechanical outcome measures associated to AJD in a young patient with advanced degeneration of the ankle joint.

## 2. Case Report

The current case report describes clinical and biomechanical outcome measures of AJD in a 14-year-old patient with severe haemophilia A. He is a cooking school student (52 kg, 1.75 m) requiring being on his feet during classes, leading to many missed classes due to ankle and foot pain. He has a history of recurrent hip and right ankle hemarthrosis from age 12 onwards. He never developed inhibitors. He developed a target joint in the right ankle after age 12 despite adequate prophylaxis (standard rFVIII 35 IU/kg, three times a week). Since January 2016, at 14, the patient had significant pain in weight-bearing situation as well as recurrent hemarthrosis and suffered from increasingly frequent ankle hemarthroses with continuous diurnal and nocturnal pain. Initially, he received conservative treatment encompassing footwear, custom made foot orthotics and physiotherapy. Walking was only possible with crutches. Despite regular prophylactic treatment and a radiosynoviorthesis with ytrium performed on 29 April 2016, the patient still presented recurrent hemarthroses with significant anterior synovial hypertrophy. After anterior open synovectomy of the right ankle performed on 14 June 2016, the mobility of the ankle was satisfactory with 15° dorsi flexion and 20° of plantar flexion. Walking was normal and monopodal jumping is possible without pain. Despite the aforementioned conservative treatment, the MRI performed on 1 June 2017 showed worsening of the osteochondritis of the talar dome with loss of convexity and sinking of the cartilage surface (Figure 2A). Subsequently, a surgical synovectomy was performed by anterior arthrotomy. As a result, he no longer sustained any ankle joint bleeds, but the ankle joint remained painful despite daily use of Cox2-selective non-steroidal inflammatory drugs (NSAID) and daily rFVIII injections.

Because of persistent and incapacitating pain and the poor response to the conservative treatments, AJD with placement of an external fixator was proposed as joint-sparing surgical intervention. We performed a preoperative functional assessment consisting of a clinical examination and quantified 3D gait analysis. The aforementioned analysis complied with a 3D multi-segment foot model protocol is described and validated extensively elsewhere [9,10]. Existing normative data of healthy age-matched subjects were included for comparison.

**Figure 1 ijerph-18-11405-f001:**
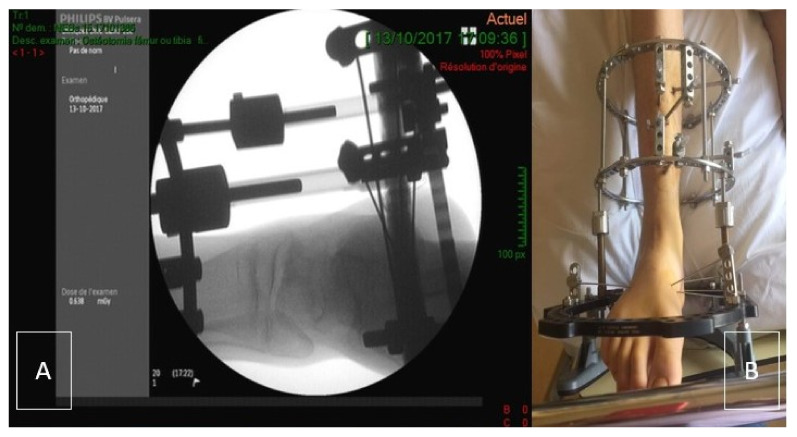
(**A**) Per-operative evaluation of Ilizarov placement using X-ray. (**B**) Clinical picture of the Ilizarov frame (post-surgical status-day 1).

**Figure 2 ijerph-18-11405-f002:**
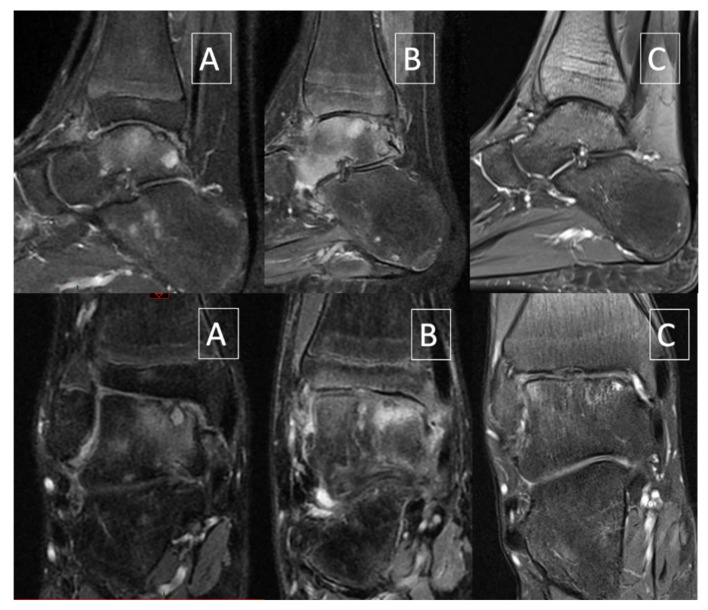
Sagittal and coronal fat-saturated proton density–weighted MRI. (**A**) June 2017 (4 months before ankle joint distraction): worsening of the cartilage lesion with collapse. (**B**) January 2018 (2 months after ankle joint distraction): edema with stable aspect of the cartilage lesion. (**C**) December 2019: decrease of edema and improvement of bony and cartilaginous lesions.

This pre-operative analysis showed lowered sagittal plane ankle range of motion, with marked dorsiflexion limitation during midstance (Figure 3). This decreased range of motion was compensated by an increased range of motion of the midfoot and a more everted position of the rearfoot. A premature internal ankle and midfoot plantarflexion moment was observed immediately after initial contact which was caused by a low-heel floor contact deviation (Figure 3). Finally, an increased power absorption was observed at the midfoot, with a peak power absorption being almost three times higher when compared to the age matched reference data. At mid- and long term, it has been hypothesized in the literature that such kinetic profiles may cause pathological joint contact forces and excessive soft tissue stress [11,12].

Approximately 12 weeks after the collapse of the talar dome was observed on MRI, the patient underwent the ankle joint distraction (AJD) procedure in October 2017. The surgical procedure was performed under general anesthesia with the patient in supine position. No tourniquet was applied. The leg was prepared with povidone-iodine (Isobetadine®, DM Amstelveen, Pays-Bas, Nederland) before sterile draping. Two Illizarov rings (155 mm-diameter) (TSF-Smith & Nephew, Memphis, TN, USA) were fixed to the tibia each with three half-pins (5 mm-diameter) (TSF-Smith & Nephew, Memphis, TN, USA) perpendicular to the long axis of the tibia. An additional foot-ring (155 mm-U-shaped; TSF-Smith & Nephew, Memphis, TN, USA) was prolonged by a half-ring (155 mm; TSF-Smith & Nephew, Memphis, TN, USA) and was placed around the foot. The foot was fixed to the U-ring with four 1.8 mm-olive-wires (TSF-Smith & Nephew, Memphis, TN, USA) (2 in the calcaneus and 2 in the mid-foot). The wires were tensioned to 40 kg. The two tibial rings and the foot ring were connected with four Ilizarov threaded shafts (TSF-Smith & Nephew, Memphis, TN, USA). Additional non-sterile rocker ring (TSF-Smith & Nephew, Memphis, TN, USA) was fixed to the U-ring at the end of the procedure to allow contact with the ground (Figure 1A,B). Clotting factors were administered perioperatively (bolus infusion followed by continuous infusion to ensure a F8 level of 80%, tapered down to 60% during the first post-operative week and to 30% during the second week). A rocker ring was added to allow the patient to have the external fixator in contact with the ground. This allowed the patient to walk without direct weight bearing of the foot. The aim was to maximally unload the joint.

Distraction began the day after the surgery and was progressive at the rate of 1 mm/a day until 5 mm distraction. Post-operative care involved daily pin site dressings for the 10 first days with povidone-iodine and later with 70° alcohol. Weight-bearing was forbidden as long as the fixator was in place. The fixator was removed in January 2018 after 12 weeks and physiotherapy started immediately after. However, not long after removal, the patient developed a complex regional pain syndrome (CRPS) with nocturnal and weight-bearing pain, erythrocyanosis, paresthesia, and dysesthesia. He was treated with daily physiotherapy, anti-inflammatory, tramadol, and intra-articular injection of hexatrione. Diagnosis of CRPS was confirmed by MRI showing extensive bone edema (Figure 2B).

From September 2018, almost 1 year after AJD, clinical improvement of pain and CRPS was observed. The patient was able to bear weight for 7 hours consecutively in school. A first post-operative 3D gait analysis revealed similar foot joint kinematic and kinetic patterns compared to baseline analysis, except for the power values of the ankle and midfoot. Between 60% and 80% of gait, the midfoot joint showed power absorption of 0.5 Watt/kg compared to a baseline value of 1 Watt/kg. At 80% of gait, the ankle joint showed a power generation of approximately 2.5 Watt/kg, compared to a baseline value of over 3 Watt/kg. Almost complete normalisation of midfoot kinetics was observed, hence, at this stage the patient demonstrated reduced ankle joint power absorption (Figure 3).

Shortly after this 1 year follow-up, the patient suffered a stress fracture of the third metatarsal bone of the right foot that healed uneventfully. In February 2019, the patient further improved and was able to walk and run without pain and MRI showed a stability of the chondral lesion (Figure 2C). Since June 2019, pain resolved completely and the patient could resume normal physical activities, including fitness three times a week. A second post-operative 3D gait analysis was performed at 18 months follow-up showing an early plantarflexion pattern at the ankle joint during midstance (Figure 3). This observation may illustrate an inhibited tibial advancement resulting in a premature heel rise during midstance. Since the clinical examination revealed a normal sagittal plane passive ankle range of motion together with a normal joint end-feeling, it was concluded that this deviated ankle joint motion pattern is part of a walking strategy in which high loading of the anterior talar joint surface is avoided. We also observed a tendency towards increased power absorption at the midfoot with peak power absorption of 1.2 Watt/kg, being almost two times higher when compared to boys of the same age. Since the latter (mal)adaptive compensations may cause overuse injuries at the foot and lower limb, the patient was advised to wear rocker-bottom footwear and custom-made foot orthotics.

More recently, in January 2020, the pain in the right ankle joint reappeared and it was decided to carry out a methylprednisolone intra-articular injection in February 2020. The latter injection made the patient pain-free until September 2020. At that time, the pain reappeared during prolonged standing. One month later, a triamcinolone acetonide intra-articular injection was administered providing satisfactory clinical benefit for two months whereafter recurrence of the painful symptoms were noted. A new MRI scan was carried out showing extensive edematous infiltration of the talus dome and the anterior slope of the tibial plafond but with a stabilisation of the cartilage lesions as regard to the MRI performed one year earlier.

## 3. Discussion

The aim of this case study was to report the short and midterm evolution of ankle joint distraction in a young and mobile haemophilic patient of 14. Present case report is the first to describe treatment of severe haemophilic ankle arthropathy with joint distraction in an adolescent patient. A previous study reported the effect of AJD in haemophilia but the described cases were three adults [5]. Furthermore, only case 1 in this study suffered from severe haemophilia, whereas in cases 2 and 3 the severity was only moderate and mild [5].

From a hematological viewpoint, it can be stated that this procedure can be performed safely in patients with severe hemophilia A, provided accurate peri-operative FVIII substitution. From a structural point of view, this AJD was a success because it firstly improved and secondly stabilised the osteo-cartilaginous lesions of the ankle. Some structural lesions persisted; however, it is reasonable to assume that further deterioration of the joint was prevented and therefore also extended the timing of end-stage surgical interventions. Even though CRPS was developed after removal, the patient further improved and was able to walk and run without pain and MRI showed a stability of the chondral lesion (Figure 2C). The three cases of Van Meegeren and colleagues also showed structural improvement on MRI, however in case 3 AJD did not cause complete disappearance of bone oedema and of some of the subchondral cysts.

Eventually, pain resolved completely and the patient could fully resume his normal physical activities. This is in contrast to the three cases of Van Meegeren and colleagues, where only a decrease in pain with 45% was reported in case 1, 70% in case 2, and 80% in case 3. Furthermore, the study of Van Meegeren and colleagues only describes ankle joint mobility, whereas this case study also describes kinematic and kinetic profiles. The addition of kinetics may provide additional insight into the pathomechanical pathways associated to AJD. It is believed that the biomechanical outcome measures associated to the 18-month follow-up are suboptimal and close monitoring of the patient is recommended. However, as this case study only captures the results of one adolescent patient with severe haemophilia A, we are well aware that this does not offer substitute for the drawing of general conclusions. Further studies on the effect of AJD in haemophilic arthropathy should be conducted.

## 4. Conclusions

Foot joint mobilisation, foot core strengthening exercises together with rocker-bottom footwear and custom-made foot orthotics are recommended as conservative treatment strategies in presence of ankle haemarthropathy. When these conservative treatments, together with radiosynoviorthesis, provide unsatisfactory results for the patient doctors are left with three surgical treatment options: fusion (arthrodesis), total ankle replacement or joint distraction. Ankle fusion provides in the majority of the cases effective pain relief, but at the expense of foot joint mobility and the risk at overloading distal joints of the foot. Total ankle replacement seems to be an interesting alternative which avoids the aforementioned limitation; however, the outcome in patients with haemophilia remains poorly studied. Taking this together, it seems logic that AJD is a valuable surgical procedure for this patient population, especially when facing young adolescents.

From a functional viewpoint, we observed a clear reduction in the patients’ physical activities until one year after AJD. Standing for a long time and walking were possible again. Short distance running was even possible without pain, even though running in itself was not a recommended physical activity due to its potential harmful stress on the cartilage. However, despite these functional and structural improvements, recurrent painful phenomena are observed and which are probably related with the development of recurrent subchondral oedema. Next to these positive observations, we would like to stress that also some negative phenomena occurred in this midterm follow-up. Next to a stress fracture of the third metatarsal it seems that the surgical intervention also triggered the development of a CRPS. Careful monitoring of the patient is therefore highly recommended in order to detect early signs of these clinical phenomena.

## Figures and Tables

**Figure 3 ijerph-18-11405-f003:**
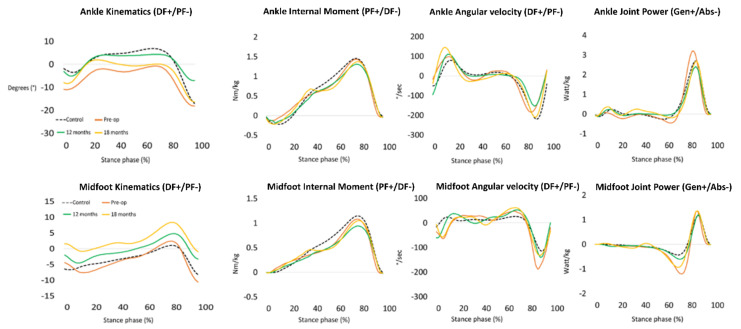
Kinematic and kinetic waveforms at pre-operative (orange line), 12 months (green line) and 18 months (yellow line) follow-up of the ankle and midfoot during stance phase of walking (X-axis). Normative values are visualized with a dashed black line.

## Data Availability

Data are available upon reasonable request.

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
