# Peer review of "Clinical and Biomechanical Progression after Ankle Joint Distraction in a Young Adolescent Patient with Haemophilia"

_ijerph, 2021, doi:10.3390/ijerph182111405_

Round 1

Reviewer 1 Report

This report highlights the short and mid-term clinical and biomechanical progression after ankle joint distraction in a young adolescent patient with hemophilia.

  1. Introduction

There is no problem because the purpose of this case report is very clear

  1. Case report

Line 79-81: In abstract, you described as “ankle joint distraction (AJD) was performed in this 14-year-old patient using an external fixator encompassing two Ilizarov full rings in the tibia and a foot ring fixed to the foot by four K-wires”. But, in the main text, surgical techniques of AJD were described rather briefly. Since this case is a report that focuses on the clinical progression after AJD, specific mensions for this procedure (AJD) must be added. (e.g., surgical instruments, technical note of this procedure, post-operative treatment and rehabilitation, etc.)

  1. Discussion

The current discussion does not sufficiently reflect the consideration of this case. It is essential to compare or analyze using previously reported related literatures review.

  1. Conclusion

Rather, it is something that should be discussed in the discussion section.

New conclusions should be stated.

Author Response

Dear reviewer

We thank you for your valuable feedback. Adjustments have been made and can be found in attached file. 

Reviewer 2 Report

Although the results provide data that may be interesting, it would be necessary to include key methodological and data that would make the study robust. 
It is recommended to include a series of key methodological data to improve understanding of the results and their clinical applicability.

Title:

  • Can be shortened to make it shorter, clearer and more concise.

Abstract:

  • Authors should include some numerical data to support the results.
  • There is no clear conclusion that the synthesis of the study offers.

Introduction

  • That a 14-year-old boy with hemophilia, who has had access to prophylactic treatment, presents a picture of advanced hemophilic arthropathy is rare. The authors should explain what this process of joint degeneration consists of, for those readers unfamiliar with hemophilic arthropathy.
  • The Introduction section does not make a true justification for the use of the ankle joint distraction. This technique, used for decades, has certain characteristics, indications and procedures that the authors do not indicate in the text.
  • The objective of the study is not stated, according to the development of the Methods

Discussion

  • The authors hardly make a discussion of the results and changes observed in the case study.

Conclusions

  • Phrases not in accordance with the objectives of the study are included.
  • The conclusions are very optimistic, based on the results obtained. If the patient has pain, which was the main pretreatment clinical manifestation, he will hardly be able to run (no matter how little) if the pain persists.

Figures

  • The use of a clinical and a radiological image of the ankle joint distraction would facilitate the visualization of professionals unfamiliar with the surgical technique.

Author Response

Dear reviewer

Thank you for your valuable feedback. Adjustments to the manuscripts have been made and a point-by-point response to your comments.

Reviewer 3 Report

This case report by Haelewijn at al is an important description of a promising treatment option for haemophilic ankle arthropathy (AJD) in young patients with the aim to diminish pain, maintain joint range of motion and functionality and postpone arthrodesis as long as possible. The structural (MRI) and biomechanical results postoperatively support the potential of AJD.

The described patient is only 14 years old and failed conservative treatment options. However, from the current description it is not clear whether shoe adaptationts and/or orthotics were also tried pre-operatively. Moreover, considering his age, a closed growth plate would be recommended before applying AJD. From the MRI I cannot fully conclude whether this was the case.

Timing of surgical interventions in haemophilic arthropathy is mainly based on clinical symptoms and functionality. It would be helpful for understanding the improvement by AJD to better describe the pre-operative status of pain intensity and functionality (post-operatively able to weight bear for 7 hours, what was the status pre-operatively?).

The authors mention twice in the manuscript that the patient was able to run post-operatively (line 124 and 167). I feel this needs a comment to clearify that even though this is possible, this should not be encouraged as it still is a damaged joint and progression will be accelerated by running.

Wieght bearing was forbidden as long as the fixator was in place. Unloading the joint is one of the hypothesized mechanisms of AJD, however in osteoarthritis, limited weight bearing during distraction is encouraged to maintain intermittent fluid pressure in the joint – maintained by using hinges, thin flexible wires, or springs in the distraction frame – at this is important for nutrition of the chondrocytes (Lafeber et al, Curr Opinion Rheumatol 2006).

In the discussion, the authors state that AJD can be performed safely, but in the case report it is not explicitely stated that the surgery was not complicated by bleeding nor infection (pin tract infection is the most common complication). Moreover, should the CRPS not be considered a complication?

Minor comment: line 135 'normal boys' - please change this, as it now implies that a patient with haemophilia is abnormal.

Author Response

Dear reviewer

Thank you for your positive and valuable feedback. Adjustments have been made and we provided a point-by-point response to your comments.

Round 2

Reviewer 1 Report

The comments when I reviewed last time has been improved well enough.